# Re-Irradiation for Head and Neck Cancer: Cumulative Dose to Organs at Risk and Late Side Effects

**DOI:** 10.3390/cancers13133173

**Published:** 2021-06-25

**Authors:** Anna Embring, Eva Onjukka, Claes Mercke, Ingmar Lax, Anders Berglund, Sara Bornedal, Berit Wennberg, Emmy Dalqvist, Signe Friesland

**Affiliations:** 1Department of Oncology, Karolinska University Hospital, 171 76 Stockholm, Sweden; clas.mercke@sll.se (C.M.); signe.friesland@sll.se (S.F.); 2Department of Oncology-Pathology, Karolinska Institute, 171 76 Stockholm, Sweden; eva.onjukka@sll.se (E.O.); ingmar.lax@sll.se (I.L.); 3Department of Medical Radiation Physics and Nuclear Medicine, Karolinska University Hospital, 171 76 Stockholm, Sweden; sara.bornedal@sll.se (S.B.); berit.wennberg@sll.se (B.W.); emmy.dalqvist@sll.se (E.D.); 4Epistat Epidemiology and Statistics Consulting, 756 55 Uppsala, Sweden; anders.berglund@epistat.se

**Keywords:** re-irradiation, cumulative dose, organs at risk, late side effects, head and neck cancer, composite DVH, carotid blowout, osteoradionecrosis

## Abstract

**Simple Summary:**

Local recurrences of head and neck cancer are unfortunately common and can be difficult to treat. The treatment is challenging, partly due to the location, with several important organs in the head and neck area, but also because recurrence often occurs in an area already treated with radiotherapy. It has been shown that repeat radiotherapy, re-irradiation, can offer long-lasting tumor control and sometimes even cure in selected patients. However, there is a risk of normal tissue close to the tumor being damaged by high cumulative doses of radiotherapy. In this study, we aim to establish levels of cumulative dose to specific organs that could be considered reasonably safe to deliver at re-irradiation without causing high rates of severe side effects. Increased knowledge in dose–response relationships in re-irradiation for head and neck cancer will facilitate a tailored treatment for the individual patient.

**Abstract:**

Re-irradiation in head and neck cancer is challenging, and cumulative dose constraints and dose/volume data are scarce. In this study, we present dose/volume data for patients re-irradiated for head and neck cancer and explore the correlations of cumulative dose to organs at risk and severe side effects. We analyzed 54 patients re-irradiated for head and neck cancer between 2011 and 2017. Organs at risk were delineated and dose/volume data were collected from cumulative treatment plans of all included patients. Receiver–operator characteristics (ROC) analysis assessed the association between dose/volume parameters and the risk of toxicity. The ROC-curve for a logistic model of carotid blowout vs. maximum doses to the carotid arteries showed AUC = 0.92 (95% CI 0.83 to 1.00) and a cut-off value of 119 Gy (sensitivity 1.00/specificity 0.89). The near-maximum dose to bones showed an association with the risk of osteoradionecrosis: AUC = 0.74 (95% CI 0.52 to 0.95) and a cut-off value of 119 Gy (sensitivity 1.00/specificity 0.52). Our analysis showed an association between cumulative dose to organs at risk and the risk of developing osteoradionecrosis and carotid blowout, and our results support the existing dose constraint for the carotid arteries of 120 Gy. The confirmation of these dose–response relationships will contribute to further improvements of re-irradiation strategies.

## 1. Introduction

Radiotherapy is a central component in the primary treatment of head and neck cancer (HNC). Unfortunately, local recurrences are common and can affect as many as 24 to 50% of patients after primary treatment [1,2,3]. Management of HNC recurrence or second primary in a previously irradiated volume is a therapeutic challenge. Surgery is typically the treatment of choice, but the low rates of tumor control have led investigators to question whether the modest benefits outweigh the potentially increased morbidity [4]. Several studies have shown that re-irradiation can offer durable local control or even cure in selected patients [5,6,7,8,9,10]. Patients with a relatively small tumor burden and longer interval (at least six months) between irradiations could be considered for re-irradiation with curative intent, whereas patients with considerable comorbidity and/or severe toxicity following prior radiotherapy are considered less suitable for such treatment [11,12]. The selection of patients suitable for re-irradiation is critical, to avoid severe side effects or even treatment-related death. Several studies have proposed different tools covering patient characteristics that are of importance in the selection [5,13,14], but little is published on cumulative doses to normal tissues and dose constraints in the re-irradiation setting. Most studies published on re-irradiation of HNC are relatively small retrospective studies and the definition of re-irradiation is often vague, making comparison between different studies difficult. Moreover, there are only a few studies presenting dose/volume data and the uncertainty regarding normal tissue tolerance in the re-irradiation setting prevails.

Factors that have been shown to increase the risk of serious side effects after re-irradiation are: higher dose at re-irradiation [15], higher dose at primary irradiation [13], concurrent chemotherapy at re-irradiation [16,17], radiation dose to specific organs, such as the carotid arteries [18] and mandible [17], shorter time between irradiations [19], prior surgery [8] and large re-treatment volumes [20]. In contrast to patient- and disease-related factors, doses and volumes at re-irradiation are to some extent modifiable. A clearer understanding of normal tissue tolerance to re-irradiation and cumulative dose constraints would provide valuable tools to tailor the individual patient’s treatment plan at re-irradiation and possibly improve the therapeutic ratio. Greater certainty regarding the risk of serious complications would also facilitate a well-informed choice of treatment strategy, in consultation with the patient.

Although the knowledge regarding dose constraints in HNC re-irradiation is generally limited, the tolerance of a few critical organs at risk (OARs) have received greater focus. An OAR that has been relatively well investigated is the spinal cord. There are published recommendations for dose constraints both in the primary setting and at re-irradiation [21,22]. Exceeding these dose constraints can result in myelopathy, a dreaded complication; fortunately, this side effect of radiotherapy is uncommon. Another dreaded and often lethal complication of HNC and/or HNC therapy is carotid blowout syndrome, in which the carotid artery or one of its major branches ruptures. Known risk factors of carotid blowout syndrome are tumor invasion of the carotid arteries, infection, surgery and high cumulative doses of radiotherapy [23]. There is no consensus regarding dose constraints for the cumulative dose to the carotid arteries. Garg et al. suggested that the risk of developing carotid blowout syndrome is higher at cumulative doses of >120 Gy [18], but this is based on a small sample with only one event. A third example of a side effect that can cause considerable suffering for the patient is osteoradionecrosis (ORN). Bots et al. conducted a retrospective analysis of 137 patients that had been treated with re-irradiation for HNC. In this study, eight patients developed ORN and one patient developed chondronecrosis of the larynx after re-irradiation, with a median cumulative dose of 114 Gy (range, 94 to 130 Gy) [17].

The aim of the current study is to generate carefully collected dose/volume data for a group of patients re-irradiated for HNC. These data will be used to explore the correlations between cumulative dose to OARs and serious side effects, in order to confirm the dose constraints for critical OARs suggested in the literature (spinal cord, carotid arteries and mandible/bony structures) and to explore additional endpoints assumed to affect the patients’ quality of life: xerostomia, dysphagia, trismus, mucosal and skin toxicity. To facilitate the interpretation of the results, a strict definition of re-irradiation has been used.

## 2. Materials and Methods

### 2.1. Patients

Fifty-four consecutive patients treated with re-irradiation for HNC in our institution between 2011 and 2017 were retrospectively analyzed. Re-irradiation was defined as having a primary treatment of ≥60 Gy and a second treatment with an overlapping volume of ≥40 Gy, resulting in a volume with a cumulative dose of ≥100 Gy (V100). Participating patients, characteristics and treatment are described in full in an earlier published article [24] and tables of patient and treatment characteristics are available in Appendix A. To facilitate comparison between different fractionation schedules, all radiotherapy doses are reported in a biologically equivalent dose of 2 Gy per fraction (EQD2) using the linear-quadratic model [25] and α/β = 3 Gy. Dose distribution summation and analysis of dose to OARs vs. side effects was carried out for all 54 included patients. The study was approved by the National Ethical Review Authority.

### 2.2. Toxicity

Grading of ORN was carried out according to Late Effects Normal Tissue Task Force Subjective, Objective, Management, and Analytic (LENT/SOMA) scores [26]. Grading of all other acute and late toxicities were according to the Radiation Oncology Group (RTOG) and the European Organization for Research and Treatment of Cancer (EORTC) Radiation Morbidity Schema [27]. Side effects were considered severe for grade ≥3. Data on toxicity were collected from a local quality registry which includes information on all patients treated with radiotherapy for HNC in our clinic. In this registry, information on acute and late toxicities from skin, mucosa, larynx, salivary glands, dysphagia, mandible, and trismus is gathered and graded accordingly. These data are collected prospectively when patients come for routine follow-up visits every three months the first two years after treatment, and then every six months for another three years. For this study, toxicity data were supplemented with a review of medical records when needed. Toxicities presenting within 90 days of the last day of re-irradiation were considered acute. Toxicities presenting after this period were considered late toxicities. The specifically investigated toxicities were myelopathy, carotid blowout, ORN, trismus and dysphagia. Other grade ≥3 late side effects (for example xerostomia and skin or mucosal ulcerations) were included in the total number of severe late side effects, but no associated OAR was considered. Carotid blowout syndrome was defined as a major bleeding from the pharynx in the absence of local recurrence.

### 2.3. Organs at Risk

The OARs investigated were the spinal cord, larynx, bones, mandible, and the carotid arteries. All OARs were delineated by a radiation oncologist specialized in HNC (AE). The spinal cord was delineated, as seen on the computed tomography (CT)-slices in level with the PTV with a minimum of 3 cm margin in cranio-caudal direction. The larynx was delineated to include the thyroid and cricoid cartilage. Bones were defined as all bony structures in the CT scan at re-irradiation. This structure was delineated using an auto-contouring tool and manually corrected in the high-dose region if needed (to exclude artefacts and non-bony structures incorrectly included). The mandible was delineated as it appears on the treatment planning images, excluding the teeth. Cranial limit was set at cranial limit of corpus mandibulae. In the carotid artery structure, the common, internal, and external carotid arteries were included bilaterally. The doses reported for the carotid arteries are the doses from the side that received the highest cumulative doses (i.e., only right or left).

The organs at risk were delineated on the CT-set from the reirradiation which was deformably registered to the CT-sets from the primary treatment for structure propagation and dose summation. Examples of the delineation of the OARs are shown in Figure 1. Dose/volume data for the spinal cord were investigated in relation to myelopathy, larynx in relation to dysphagia and laryngeal side effects (hoarseness, edema, chondritis and necrosis), bones/mandible in relation to ORN and carotid arteries in relation to carotid blowout syndrome.

### 2.4. Dose Accumulation

The 3 D dose distribution was collected from each course of treatment, including external beam radiotherapy and brachytherapy as described in our previous article [24]. For each patient, the different treatment planning images were registered to the most recent image set, which served as a reference image. A deformable registration was performed in Raystation using a mutual-information-based B-spline algorithm. The registration was visually validated for all patients, and a quantitative validation was performed for a subgroup of 14 patients. The latter consisted of manual delineation of three OARs (larynx, mandible, and carotids) in all treatment-planning images, for each patient, followed by a comparison of dose/volume statistics associated with the propagated OAR from deformable registration and the reference image structures, respectively.

The planned dose distribution from each course of treatment was converted to EQD2, using an in-house tool. This was followed by the dose accumulation from all courses of treatment on the reference CT series, facilitated by the deformable registration. Accumulated dose/volume statistics in EQD2 were then extracted for the OARs for each patient. 

### 2.5. Statistics

The time to event approach using the Kaplan–Meier method was performed for each toxicity event separately. The adverse events were classified into acute or late events depending on time from re-radiation (within 90 days or more than that). In a subsequent step, a *t*-test and univariate logistic regression models were used to study potential associations between independent variables (tumor site, time between irradiations, re-irradiated volume (V100), PTV at re-irradiation) and toxicity. Receiver–operator characteristics (ROC) curves were used for the performance between dose/volume parameters and toxicity and to determine any cut-off values by using the Youden Index (J) to use as dose constraints. ROC analysis was performed for OAR with correlating events of severe toxicity to try and establish dose constraints (carotid arteries, bones, larynx). Statistical significance was considered at a 5% alpha level and all the analysis was performed using R version 3.5.1.

## 3. Results

### 3.1. Patient and Treatment Characteristics

As described in our previous work [24], a total of 54 patients were included in the analysis and the overall survival at two and five years was 42.6 and 27.3%, respectively, and progression free survival at two and five years was 32.5 and 28.5%, respectively. The median follow-up time after re-irradiation was 54.1 months (range 34.3 to 66.3) in surviving patients and 20.1 months (range 0 to 69.9) in all included patients. The majority (94%) of the included patients had a good performance status (ECOG 0–1) at the start of re-irradiation. Definitive re-irradiation (59%) was more common compared to post-operative re-irradiation (41%). The median re-irradiation dose was 59 Gy and the most common tumor site was the oropharynx (33%) followed by cancer of the oral cavity (31%).

### 3.2. Toxicity

The overall rate of any event of severe (grade ≥3) acute and late toxicity was 26% and 51%, respectively. There were two cases of fatal carotid blowout and these events occurred 15 and 38 months after re-irradiation. No cases of radiation-induced myelopathy were observed. Multiple (>1) severe late side effects were seen in nine out of the 19 (47%) patients that experienced serious late side effects. Seven patients died within 90 days of completing the reirradiation and are not eligible for analysis of late side effects. Most severe toxicity developed within 1.5 to 2 years after re-treatment, but ORN, dysphagia and carotid blowout appeared even after two to three years (Figure 2). The median time to develop any severe late side effect was 17 months. At closure of the database (20 December 2018) 11 out of 54 patients (20%) were alive without disease, and at two years after re-irradiation, seven (64%) of these patients had experienced no severe late side effects (Table 1).

Due to only one out of four events of ORN presenting in the mandible, no correlations between doses to the mandible and ORN were found and further dosimetric analysis considering ORN were on doses to bones.

### 3.3. Predictors of Toxicity

We found no significant correlation between tumor site, time between irradiations, re-irradiated volume (V100), re-irradiation volume (PTV at re-irradiation) or severe side effects at first irradiation and the risk of developing any severe late side effects. The characteristics of all patients that experienced one or more grade ≥3 late side effect are displayed in Table 2. The cumulative median near-maximum dose (D1cc) to the spinal cord, carotid arteries, larynx, and bones were 34 Gy (range 16 to 52 Gy), 97 Gy (range 47 to 139 Gy), 70 Gy (range 0 to 130 Gy) and 118 Gy (range 69 to 141 Gy), respectively. The cumulative median mean dose to the larynx was 46 Gy (range 0 to 79 Gy). The doses to OARs for all patients are displayed in Figure 3a–d, highlighting patients with serious toxicity after re-treatment; the time between irradiations and the time to follow-up are indicated for each patient. Neither the time between irradiations nor the follow-up has any apparent influence on the outcome. 

The composite dose volume histograms (DVH) show maximum doses to the carotid arteries to be near or above the upper third quartile for the two patients experiencing carotid blowout, implying higher cumulative maximum doses in these patients (Figure 4a). ROC-curve for a logistic model of carotid blowout vs. maximum doses to the carotid arteries showed AUC = 0.92 (95% CI 0.83 to 1.00) and a cut-off value of 119 Gy (sensitivity 1.00/specificity 0.89) (Figure 5a). The near-maximum dose to bones showed a significant association with the risk of ORN: AUC = 0.74 (95% CI 0.52 to 0.95) and a cut-off value of 119 Gy (sensitivity 1.00/specificity 0.52) (Figure 5b); although with less separation between patients with and without toxicity compared to carotid blowout. This association can also be seen in the composite DVH for bones (Figure 4b). In contrast, the ROC analysis showed no significant association between the mean larynx dose and the risk of dysphagia (Figure 5c); this is similarly illustrated in Figure 3c and Figure 4c, where the patients with serious dysphagia did not generally receive higher dose to the larynx.

## 4. Discussion

This is one of the larger studies on re-irradiation for HNC that presents extensive dose/volume data, cumulative doses to OARs and side effects for all included patients. Our strict definition of re-irradiation may have contributed to the seemingly high rates of late side effects. Previously published studies with a vaguer definition of re-irradiation may have included patients with an overlap only in low dose volumes, resulting in an underestimation of the incidence of side effects after re-irradiation. Another factor that may have contributed to the seemingly high rates of late side effects is the relatively long time to last follow-up of 54 months in surviving patients. We specifically explored the side effects carotid blowout syndrome, dysphagia, ORN and trismus, but all grade ≥3 side effects were recorded, thus considering the most common side effects after re-irradiation [20].

The cut-off value of 119 Gy (EQD2) for the maximum dose to the carotid arteries was very similar to the cut-off value of 120 Gy suggested by Garg et al. [18] (Table 3). This is a valuable validation of these results, although this is a dose constraint that will not always be possible to follow, as the carotid arteries are quite often in close proximity of both the prior high dose volume and the planned re-irradiation volume. As standard primary treatment is often 68 to 70 Gy [28,29] and the recommended re-irradiation dose is 60 to 68 Gy [30,31], an overlap of these two volumes close to the carotid artery will result in a cumulative dose well over 119 to 120 Gy, so to stand a chance of eradicating the tumor, the dose constraint will have to be exceeded. However, the knowledge of an increased risk of carotid blowout syndrome, if exceeding the dose limit, is valuable in the discussion with the patient on deciding what treatment option to choose and in the discussion on risk–benefit balancing in re-irradiation.

In the previously mentioned study by Bots et al., patients that developed ORN after re-irradiation had received a median cumulative dose of 114 Gy (physical dose to site of complication) [17] (Table 3). This is not far from the cut-off value of 119 Gy to bones we found in our material. This value should be more reliable compared to Bots’ study, where cumulative dose distributions were not produced, but an estimation of the maximum summed radiation point dose in the OARs was made. This was due to the lack of volume data from patients treated in earlier years and it was therefore not possible to analyze dose/volume relationships, a weakness pointed out by the authors themselves. Furthermore, Bots stated that there were no cases of mandibular ORN in the group of patients with a cumulative dose of <100 Gy, which corresponds well with our results of no cases of ORN in the group of patients with a cumulative dose of <100 Gy to bones.

In contrast to previously published data, we found no significant correlation between the time between irradiations or re-irradiation volume (PTV at re-irradiation) and the risk of developing any severe late side effects. In the study by Lee at el., shorter time (<20 months) between irradiations and a PTV > 100 cm^3^ were both independent predictors of developing severe late side effects [19]. Additionally, Phan et al. have shown that a larger treatment volume at re-irradiation (clinical target volume > 50 cm^3^) was correlated to both acute and severe late side effects [32]. The reason these correlations were not demonstrated in our study could be our relatively small cohort.

In our study, the most severe late toxicity developed within 1.5 to 2 years after re-treatment and the median time to develop any severe late side effect was 17 months. This is consistent with the findings in the study by Ward et al. [13]. In that large multi-institutional study including 505 patients, severe late side effects presented at a median of 9.2 months and the majority of all late side effects presented within two years (74 of 85 events). Multiple severe late side effects were seen in 21% of the patients compared to 47% in our study. A contributing factor to the Ward study showing a median time to event shorter than ours and considerably less multiple severe late side effects could be our longer follow-up time (54.1 months compared to Ward’s 21.5 months, in surviving patients) and therefore the recording of more events occurring later.

It is difficult to value the incidence of side effects after re-irradiation and what rates could be considered acceptable in the cost–benefit balance between toxicity and tumor control. To explore this further, we compared the incidence of dysphagia and trismus in this re-irradiation cohort with the incidence of the same side effects in a reference cohort of more than 700 patients treated with primary radiotherapy with curative intent. A detailed description of this evaluation is given in Appendix B. 

This comparison showed that the cumulative incidence of dysphagia (grade ≥3) in an identified subgroup (high laryngeal dose) of 17% of the patients in the reference cohort had about the same cumulative incidence of dysphagia as the patients that have received re-irradiation (24% at three years) (see Appendix B, Figure A1a,b). This could imply that the incidence of dysphagia in the re-irradiation cohort is not excessive. 

The incidence of trismus (grade ≥3) was much higher in the re-irradiated cohort compared to the reference cohort (see Appendix B, Figure A1c,d). This could be due to the extensive treatment many of the patients with recurrent HNC have experienced compared to patients treated for primary HNC only. However, the cumulative incidence of trismus in the re-irradiation cohort was still reasonably low (12% at three years) and could be considered acceptable due to the clinical situation.

Patient-related factors, such as smoking status and comorbidity, have been shown to have an impact on the development of side effects after radiotherapy in general and re-irradiation in particular [14,33], and as previously mentioned, the growth pattern of the tumor in relation to the carotid arteries may play a role in the risk of developing carotid blowout syndrome [23]. A weakness of our study is that these patients and tumor-related factors were not available, likely confounding our results to some degree.

When considering re-irradiation for HNC patients in our clinic, we follow our local guidelines for patient selection and treatment, based on the current literature. Our only strict dose constraint is for the cumulative dose to the spinal cord (and where relevant, also the brainstem), which is more highly prioritized than covering the target volume with the prescribed dose. For the spinal cord, we assume some normal tissue repair between the irradiations, as suggested by Nieder et al. [22]. A dose constraint of lower priority is the cumulative dose to the carotid arteries, where we avoid doses >120 Gy when possible, but covering the target volume is of a higher priority, which sometimes results in exceeding this constraint. Doses to other OARs and overlapping re-irradiation volumes are evaluated individually for each patient. 

As Garg [18] and Bots [17] have stated before us, cumulative dose/volume data are essential to improve re-irradiation strategies. Radiotherapy requires a fine balance between sufficiently high doses to the tumor, while maintaining doses to OARs at an acceptable level. At re-irradiation, higher rates of severe side effects are expected compared to after primary treatment, due to the higher cumulative doses. This can to some extent be considered acceptable, because of the limited treatment options available for patients with recurrent or second primary HNC. Re-irradiation can offer long-term local control and sometimes even cure, but this might come with a high level of severe side effects. However, the current study shows that not all patients re-irradiated for HNC suffer from severe late side effects. In the 11 patients without disease at closure of our data base, the majority (64%) had experienced no severe late side effects two years after re-irradiation. Greater knowledge regarding normal tissue tolerance at re-irradiation will facilitate improved treatment plans and might improve the risk–benefit ratio. Alternatives in immunotherapy are also emerging. Immunotherapy alone will only help a small proportion of patients with recurrent HNC, but a combination of different modalities, such as re-irradiation and immunotherapy, will possibly result in better outcomes. There are currently several trials registered on clinicaltrials.gov investigating this combination. Further studies on re-irradiation for HNC are needed to explore dose/volume data and new combinations of modalities, and preferably, these studies would also include quality of life data.

## 5. Conclusions

Careful accumulation of dose to OARs demonstrated an association between cumulative dose and the risk of developing ORN and carotid blowout, and the results support the existing dose constraint for the carotid arteries of 120 Gy. Re-irradiation should be considered as a treatment option for patients with recurrent HNC as it has been shown to offer long-lasting tumor control and even cure for selected patients and this material shows that a group of patients will have long-lasting tumor control without experiencing severe late side effects. With careful consideration of cumulative dose/volume data, treatment plans can be tailored to optimize the risk–benefit ratio for the individual patient.

## Figures and Tables

**Figure 1 cancers-13-03173-f001:**
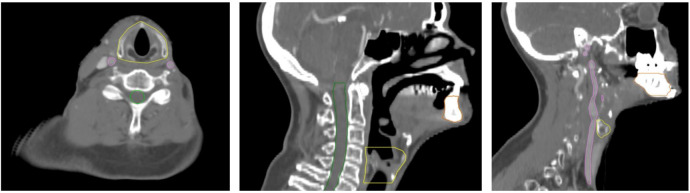
Delineation of organs at risk: larynx (yellow), spinal cord (green), mandible (brown), carotid artery (pink).

**Figure 2 cancers-13-03173-f002:**
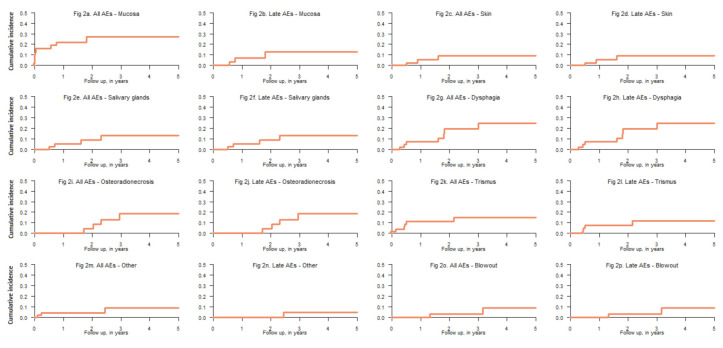
Time to event for all investigated severe side effects. Plots showing both time to any severe side effect and time to severe late side effect. AE—adverse event, grade ≥3 side effect. Others—one patient with grade 5 acute radiation-induced toxicity, one patient with neuropathy of the hypoglossal nerve, one patient with aspiration pneumonia.

**Figure 3 cancers-13-03173-f003:**
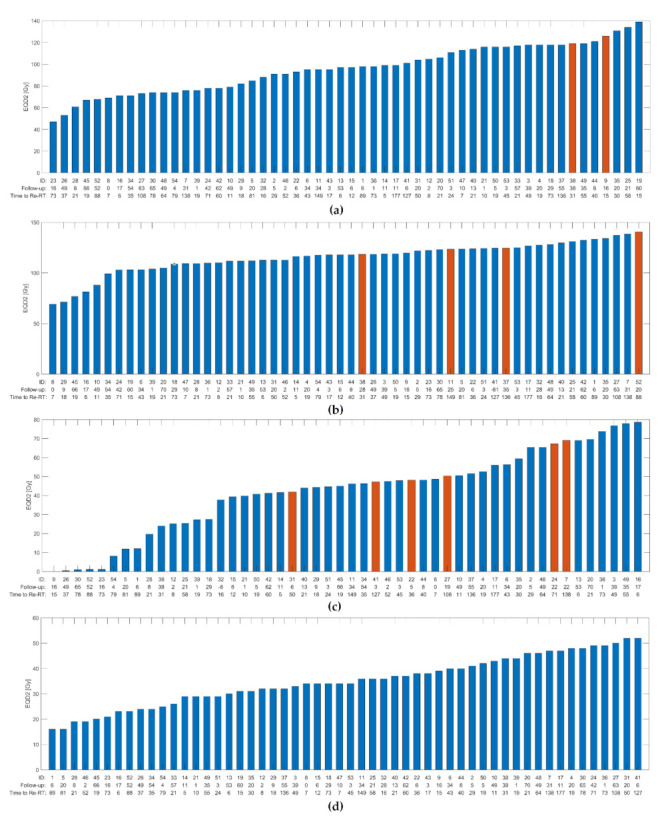
Histograms of cumulative dose in EQD2 in all included patients. Red bars indicating patient with severe side effect. The time between irradiations and the time to follow-up are indicated for each patient. (**a**) Maximum dose (D1cc) to carotid artery and carotid blowout syndrome. (**b**) Maximum dose (D1cc) to bones and grade ≥3 osteoradionecrosis. (**c**) Mean dose to larynx and grade ≥3 dysphagia. (**d**) Maximum dose (D1cc) to spinal cord and no events of neuropathy. Abbreviations: ID—identification number of subject, Re-RT—re-irradiation.

**Figure 4 cancers-13-03173-f004:**
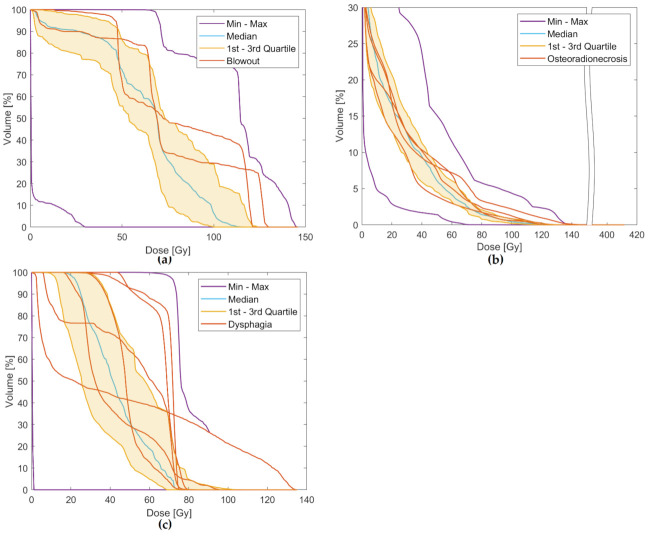
Composite dose volume histograms (DVH) highlighting the individual DVHs of patients with severe side effects. (**a**) Dose in EQD2 to carotid artery and blowout syndrome. (**b**) Dose in EQD2 to bones and grade ≥3 osteoradionecrosis. (**c**) Dose in EQD2 to larynx and grade ≥3 dysphagia.

**Figure 5 cancers-13-03173-f005:**
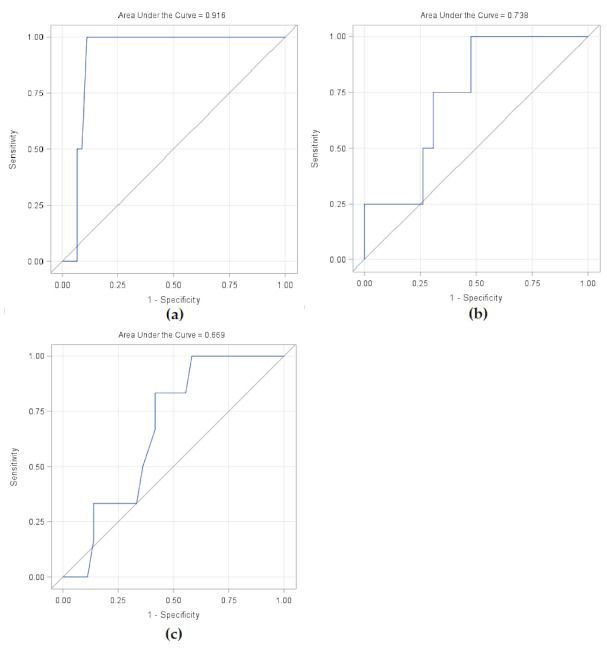
Receiver–operator characteristics (ROC) curve for logistic model. (**a**) Maximum dose (D1cc) to carotid artery and carotid blowout syndrome. AUC 0.92 (95% CI 0.83 to 1.00) and a cut-off value of 119 Gy (sensitivity 1.00/specificity 0.89). (**b**) Maximum dose (D1cc) to bones and grade ≥3 osteoradionecrosis. AUC = 0.74 (95% CI 0.52 to 0.95) and a cut-off value of 119 Gy (sensitivity 1.00/specificity 0.52). (**c**) Mean dose to larynx and grade ≥3 dysphagia. AUC = 0.67 (95% CI 0.49 to 0.85).

**Table 1 cancers-13-03173-t001:** Severe late side effects at two years in patients without disease at closure of database.

Subject	Late Side Effects Grade ≥3 at Two Years After Re-Irradiation
10	None
11	Mucosa, Osteoradionecrosis
13	None
19	None
26	None
27	Skin, Mucosa, Xerostomia, Dysphagia
35	None
37	None
42	Xerostomia
45	None
52	Osteoradionecrosis, Trismus

**Table 2 cancers-13-03173-t002:** Table of characteristics of all patients with one or more grade ≥3 late side effect(s).

Subject	Tumor Site at First Presentation	Late Side Effects Grade ≥3 after Re-RT									Surgery at First Presentation	Surgery Before Re-RT	PS at Re-RT	Grade ≥3 side effect at 1 st RT	Time between RT and re-RT (months)	Age at Re-RT (years)	PTV at Re-RT (cm³)	V100 at Re-RT (cm³)	Concomitant Systemic Therapy at Re-RT
		Skin	Mucosa	Larynx	Salivary Glands	Dysphagia	ORN	Trismus	Blowout	Other									
52	Sino/nasal	No	No	No	No	No	Grade 4	Grade 3	No	No	Primary tumor	No	0	No	88	41	62	71	No
42	Nasopharyngeal	No	No	No	Grade 3	No	No	No	No	Grade 3	No	No	0	No	60	55	248	91	No
23	Oral cavity	No	Grade 3	No	No	No	No	No	No	No	Primary tumor	No	1	No	73	69	262	119	No
27	Oral cavity	Grade 3	Grade 3	No	Grade 3	Grade 3	No	No	No	No	Primary tumor	Primary tumor	0	No	108	62	198	256	No
30	Oral cavity	No	No	No	No	No	No	Grade 3	No	No	No	No	1	No	78	60	161	167	No
41	Oral cavity	No	No	No	No	Grade 3	No	No	No	No	Primary tumor + ND	No	0	Yes	127	60	44	82	No
7	Oropharyngeal ^†^	No	Grade 3	No	Grade 3	Grade 3	No	No	No	No	No	Primary tumor	0	No	138	74	309	87	No
17	Oropharyngeal ^†^	No	No	No	No	No	No	Grade 3	No	Grade 3	No	No	1	Yes	177	71	102	141	No
22	Oropharyngeal *	No	No	No	No	Grade 4	No	Grade 4	No	No	No	No	0	No	36	68	128	142	No
31	Oropharyngeal *	Grade 3	No	No	Grade 3	Grade 3	No	Grade 3	No	No	No	Primary tumor	0	No	50	61	333	120	No
37	Oropharyngeal *	No	No	No	No	No	Grade 4	No	No	No	No	Primary tumor	0	No	136	59	219	154	No
38	Oropharyngeal *	No	No	No	No	No	Grade 3	No	Grade 5	No	No	Primary tumor	1	No	31	56	106	138	No
6	Hypopharyngeal	No	Grade 4	No	No	No	No	No	No	No	No	No	0	No	43	74	93	64	No
20	Hypopharyngeal	No	Grade 3	No	No	No	No	No	No	No	No	No	0	Yes	21	64	149	109	No
24	Hypopharyngeal	No	No	No	No	Grade 3	No	No	No	No	No	Primary tumor + ND	0	No	71	64	26	56	No
9	Larynx	No	No	No	No	No	No	No	Grade 5	No	Primary tumor + ND	ND	1	No	15	74	66	92	No
19	Larynx	No	No	No	No	Grade 4	No	No	No	No	No	Primary tumor + ND	0	No	15	68	452	76	No
5	Salivary gland	Grade 3	No	No	No	No	No	No	No	No	Primary tumor + ND	No	0	No	81	77	283	89	No
11	Unknown primary	No	Grade 4	No	No	No	Grade 3	No	No	No	ND	Primary tumor	0	No	149	76	154	166	No
Median (whole cohort)													95% was PS 0–1		36	63	145	90	

Abbreviations: Re-RT—re-irradiation, ORN—osteoradionecrosis, ND—neck dissection, PS—Eastern Cooperative Oncology Group Scale of Performance status, RT—radiotherapy, PTV—planning target volume, V100—volume with a cumulative dose of ≥100 Gy, HPV—human papilloma virus. * HPV positive, ^†^ HPV status unknown.

**Table 3 cancers-13-03173-t003:** Table of dose/volume predictors of toxicity after re-irradiation.

Author, Year, Reference Number	Cumulative Maximum Dose *	
	Carotid Blowout	Osteoradionecrosis
Embring et al., 2021(current article)	119 Gy	119 Gy
Garg et al., 2016 [18]	120 Gy	-
Bots et al., 2017 [17]	-	114 Gy

* maximum dose or near-maximum dose.

## Data Availability

Research data are stored in an institutional repository and will be shared upon reasonable request to the corresponding author.

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
