# Peer review of "Re-Irradiation for Head and Neck Cancer: Cumulative Dose to Organs at Risk and Late Side Effects"

_cancers, 2021, doi:10.3390/cancers13133173_

Round 1
Reviewer 1 Report
Dear Sirs,
The article is about the Re-irradiation for HNC in case of progression of the disease and concentrates of the dose and side effects of the re-irradiation.
Personally, I think it is well structured and well written. It will certainly add significantly to the pertinent literature.
As the topic is interesting not only for radiation oncologists but for medical oncologists and surgeons as well, it would be beneficial if there were some additional information being useful for multidisciplinary board discussions. Therefore, I would suggest the authors to add some lines in the Introduction about the indications and contraindications, i.e. what to take into account when considering Re-irradiation. This was clearly presented in the following articles:
When is re-irradiation in head and neck squamous cell carcinoma not indicated? Eur Arch Otorhinolaryngol 2014
Recurrent and second primary squamous cell carcinoma of the head and neck: when and how to reirradiate Head Neck 2015
I think the article is very good and am warmly suggesting to publish it.
Author Response
We thank Reviewer 1 for the critical comments and helpful suggestions. We have taken these comments and suggestions into account, and they have improved our manuscript considerably.
I would suggest the authors to add some lines in the Introduction about the indications and contraindications.
- We have added a few lines in the Introduction about what to take into account when considering re-irradiation, using the valuable references suggested.
Reviewer 2 Report
This is an interesting manuscript in which the authors evaluate the effects of head and neck re-irradiation, specifically focusing on dose-response relationships with severe late effects. While the topic is not necessarily novel, it is important work and, combined with other similar publications, helps to inform practicing radiation oncologists who are treating these patients. I do have a few minor comments to the authors which are as follows.
One of the shortcomings with this work, and inherent in most, if not all, publications on the topic, is the relatively small number of patients with longitudinal follow-up and the limited number of events. The conclusions that the authors reach on dose constraints, though, do agree with several previously published manuscripts on late effects of head and neck re-irradiation as they aptly discuss in the paper. It may be helpful for the authors to also include a table listing the toxicities/calculated dose constraints identified by this work along with those published by other groups. This would allow the reader to quickly summarize how the findings of the present study compare with prior work and potentially make the manuscript more impactful to the average reader.
Regarding the patients who did develop severe effects such as carotid blowout, was any analysis performed to determine whether tumor or patient factors may have contributed to the development of these toxicities? Some prior groups have suggested that tumor involvement, namely circumferential encasement of the carotid, is a better predictor of carotid blowout risks following re-irradiation compared with dosimetry data alone. Similarly diabetes and cigarette smoking are both microvascular insults which have been shown to increase the risk of ORN after head and neck radiotherapy.
Author Response
We thank Reviewer 2 for the critical comments and helpful suggestions. We have taken these comments and suggestions into account, and they have improved our manuscript considerably.
It may be helpful for the authors to also include a table listing the toxicities/calculated dose constraints identified by this work along with those published by other groups.
- We have added a table (Table 3) of Dose/volume predictors of toxicity after re-irradiation, to illustrate our findings more clear and facilitate a comparison with previously published data.
Regarding the patients who did develop severe effects such as carotid blowout, was any analysis performed to determine whether tumor or patient factors may have contributed to the development of these toxicities?
- Unfortunately, we do not have information on smoking status, co-morbidity (such as diabetes) and growth pattern of the tumour in correlation to the carotid arteries for all included patients in this cohort. These are all important factors that may well have influenced the outcome. This is a weakness of our study, and this has now been pointed out in the Discussion. In this study we defined carotid blowout syndrome as a major bleeding from the pharynx in absence of local recurrence, so major bleedings occurring with persistent tumour on site were not counted as side effects. But we have no knowledge in whether an eradicated tumour may have caused damage to the carotid artery walls and in this way contributed to a major bleeding.